# Obeticholic Acid Derivative, T-2054 Suppresses Osteoarthritis via Inhibiting NF-κB-Signaling Pathway

**DOI:** 10.3390/ijms22083807

**Published:** 2021-04-07

**Authors:** Dandan Guo, Liming He, Yaoxin Gao, Chenxu Jin, Haizhen Lin, Li Zhang, Liting Wang, Ying Zhou, Jie Yao, Yixin Duan, Renzheng Yang, Wenwei Qiu, Wenzheng Jiang

**Affiliations:** 1Department of Shanghai Key Laboratory of Regulatory Biology, East China Normal University, Shanghai 200241, China; 52181300030@stu.ecnu.edu.cn (D.G.); 52181300027@stu.ecnu.edu.cn (Y.G.); 51201300131@stu.ecnu.edu.cn (C.J.); 51191300110@stu.ecnu.edu.cn (H.L.); 52171300027@stu.ecnu.edu.cn (L.Z.); 51181300130@stu.ecnu.edu.cn (Y.Z.); 52201300031@stu.ecnu.edu.cn (J.Y.); 51191300078@stu.ecnu.edu.cn (Y.D.); 51201300080@stu.ecnu.edu.cn (R.Y.); 2Shanghai Engineering Research Center of Molecular Therapeutics and New Drug Development, School of Chemistry and Molecular Engineering, East China Normal University, 3663 North Zhongshan Road, Shanghai 200062, China; 51194300067@stu.ecnu.edu.cn (L.H.); 51164300228@stu.ecnu.edu.cn (L.W.)

**Keywords:** T-2054, osteoarthritis, RAW 264.7 cell line, ATDC5 chondrocytes, NF-κB-signaling

## Abstract

Osteoarthritis (OA), a degenerative joint disorder, has been reported as the most common cause of disability worldwide. The production of inflammatory cytokines is the main factor in OA. Previous studies have been reported that obeticholic acid (OCA) and OCA derivatives inhibited the release of proinflammatory cytokines in acute liver failure, but they have not been studied in the progression of OA. In our study, we screened our small synthetic library of OCA derivatives and found T-2054 had anti-inflammatory properties. Meanwhile, the proliferation of RAW 264.7 cells and ATDC5 cells were not affected by T-2054. T-2054 treatment significantly relieved the release of NO, as well as mRNA and protein expression levels of inflammatory cytokines (IL-6, IL-8 and TNF-α) in LPS-induced RAW 264.7 cells. Moreover, T-2054 promoted extracellular matrix (ECM) synthesis in TNF-α-treated ATDC5 chondrocytes. Moreover, T-2054 could relieve the infiltration of inflammatory cells and degeneration of the cartilage matrix and decrease the levels of serum IL-6, IL-8 and TNF-α in DMM-induced C57BL/6 mice models. At the same time, T-2054 showed no obvious toxicity to mice. Mechanistically, T-2054 decreased the extent of p-p65 expression in LPS-induced RAW 264.7 cells and TNF-α-treated ATDC5 chondrocytes. In summary, we showed for the first time that T-2054 effectively reduced the release of inflammatory mediators, as well as promoted extracellular matrix (ECM) synthesis via the NF-κB-signaling pathway. Our findings support the potential use of T-2054 as an effective therapeutic agent for the treatment of OA.

## 1. Introduction

Osteoarthritis (OA) is a degenerative joint disorder with main pathological features of progressive subchondral bone sclerosis, articular cartilage, osteophyte formation and synovial inflammation and has been reported as the most leading cause of disability worldwide. Osteoarthritis is more common in people over the age of 60. There is a grand challenge to treat OA [1,2,3,4,5].

OA’s progress is related to many factors, such as an imbalance of oxidant-antioxidant levels and the production of inflammatory cytokines. Production of inflammatory factors (TNF-α, IL-8 and IL-6) plays an important role in the pathogenesis, which caused arthritic pain and putative disease progression. Therefore, reducing the production of inflammatory cytokines has been regarded as a valid strategy for treating OA [1]. Until now, the treatment of OA remains a major challenge [5,6]. The inducible nitric oxide synthase (iNOS) has been demonstrated to be one of the key inflammatory mediators, participates in the release of nitric oxide (NO) and binds to cyclooxygenase-2 (COX-2) to regulate inflammatory response [7,8]. Articular cartilage injury is the main factor leading to developing OA. The pathological features of articular cartilage injury include chondrocyte apoptosis and a decrease in the ability to synthesize extracellular matrix (ECM), which mediated catabolic cytokines by increasing the expression of matrix metalloproteinases (MMPs) and ADAMTS [9,10,11,12].

Nuclear factor kappa-B (NF-κB) signaling pathway, a transcription factor, is directly activated, which results in the production of cytokines (TNF-α, IL-8 and IL-6) and NO in OA [13]. Moreover, the NF-κB-signaling pathway plays an important role in articular cartilage damage by regulating hypoxia-inducible factor-2α (HIF-2α) transcriptional activity [14]. The phosphorylation of NF-κB p65 is the key element in the activation of the NF-κB-signaling pathway. Many studies report that blocking the NF-κB-signaling pathway could attenuate developing OA [13].

In recent years, a large number of new pathology [4,15]. It has been reported that low approaches have been explored to treat OA. For example, anti-inflammatory drugs, Traditional Chinese medicine (TCM) or mesenchymal stem cell therapies were applied to relieve pain, and slow progression doses of methotrexate (MTX) may benefit OA through various mechanisms [16]. However, the adverse effects, cost and potential risks of these therapies need to be further investigated.

Obeticholic acid (OCA (6-ethyl-chenodeoxycholic acid)), a semisynthetic bile acid analog, binds with a high affinity for the farnesoid X receptor (FXR) to suppress lipogenesis and reduce de novo cholesterol biosynthesis [17]. Moreover, OCA relieved insulin resistance in an obese model and protected rats from gut barrier dysfunction in a cholestasis model [18,19]. It is reported that OCA and OCA derivatives inhibited the expression of hepatic proinflammatory cytokines in GalN/LPS-induced acute liver failure [20]. However, the relationship between OCA derivatives and OA is still not clear.

Herein, we reported the discovery of an anti-osteoarthritis lead compound by screening our small synthetic library of OCA derivatives. In the present study, we investigated the role of OCA and OCA derivatives T-2035, T-2036, T-2055-2, T-2056, T-2054, T-2052, T-2005, T-2055-1 in LPS-induced RAW 264.7 cells and TNF-α-induced ATDC5 cells. Our data showed that T-2054 effectively suppressed the expression of iNOS, COX-2 and the release of inflammatory cytokine, as well as promoted extracellular matrix (ECM) synthesis via the NF-κB-signaling pathway.

## 2. Results

### 2.1. Inhibitory Activity on LPS-Induced NO Release

Structures and synthesis of OCA and OCA derivatives were showed (Figure 1a, Appendix A). Nitric oxide (NO), an inflammatory mediator, is highly produced in response to inflammatory stimuli [21]. The release of NO was tested to investigate the anti-inflammatory effect in LPS-induced RAW 264.7 cells. Among the tested compounds, T-2054 observably inhibited LPS-induced NO release (IC_50_ value: 0.484 ± 0.127 μM) in RAW 264.7 cells compared with the LPS group (Figure 1b,c).

### 2.2. Effect of T-2054 on Cell Survival in RAW 264.7 Cells and ATDC5 Cells

Before exploring the therapeutic effects of T-2054, the cell viability was tested by the CCK-8 kit to explore the potential cytotoxicity on RAW 264.7 cells and ATDC5 cells. Our data showed no obvious cytotoxicity was observed after T-2054 treatment at different concentrations (0, 0.1, 1, 10, 100 μM) in both RAW 264.7 cells and ATDC5 cells. Thus, the optimal concentration was chosen in subsequent experiment processes (Figure 2a–c).

### 2.3. T-2054 Inhibited the Expression of IL-6, IL-8 and TNF-α in LPS-Induced RAW 264.7 Cells

The production of inflammatory cytokines is the main cause of OA and leads to the activation of various signaling pathways to aggravate the symptoms [22,23,24]. The inhibitory effects of T-2054 on gene and protein levels of IL-6, IL-8 and TNF-α were determined by real-time quantitative PCR and enzyme-linked immunosorbent assay (ELISA). It was shown that the release of inflammatory cytokines was obviously elevated by lipopolysaccharides (LPS). On the contrary, T-2054 suppressed the expression of these inflammatory cytokines. These results showed that T-2054 certainly attenuated inflammatory response in LPS-activated RAW 264.7 cells (Figure 3a–f).

### 2.4. T-2054 Inhibited the Expression of SOX9, MMP9 and ADAMTS5 in TNF-α-Induced ATDC5 Cells

Chondrocyte apoptosis, aging and the reduction of extracellular matrix (ECM) synthesis are the main pathological features, which are regulated by SOX9, MMPs and ADAMTS families [11,12]. We found that the expression of SOX9 was significantly downregulated, and MMP9 and ADAMTS5 were obviously upregulated. However, T-2054 increased the expression of SOX9 and suppressed the expression of MMP9 and ADAMTS5 in TNF-α-induced ATDC5 cells. These results showed that T-2054 promoted the extracellular matrix (ECM) synthesis (Figure 4a–c).

### 2.5. T-2054 Inhibited LPS-Induced Expression of COX-2 and iNOS in LPS-Induced RAW 264.7 Cells

Production/activity of inducible nitric oxide synthase (iNOS) and cyclooxygenase-2 (COX-2) play an important role in the inflammatory response [25]. Considering the role of iNOS and COX-2 in inflammation, the expression of iNOS and COX-2 were analyzed by Western blot. As expected, the expression of iNOS and COX-2 were markedly increased after stimulating with LPS. However, T-2054 significantly reduced the expression of iNOS and COX-2, which indicated the treatment of T-2054 could affect the expression of iNOS and COX-2 in LPS-induced RAW 264.7 cells (Figure 5a–c).

### 2.6. T-2054 Inhibited NF-κB Signaling Pathway in LPS-Induced RAW 264.7 Cells and TNF-α-Induced ATDC5 Cells

Emerging studies reveal that inflammation is closely related to the NF-κB-signaling pathway, which binds to the target gene, facilitates the transcription of downstream inflammatory mediators and associates with developing articular cartilage damage in OA [26,27]. To investigate the anti-inflammatory mechanism of NF-κB, the phosphorylation level of NF-κB p65 was determined using Western blot in LPS-activated RAW 264.7 cells and TNF-α-induced ATDC5 cells. Our data showed that T-2054 significantly reduced the phosphorylation of NF-κB p65 in LPS-induced RAW 264.7 cells. At the same time, similar results were found in TNF-α-induced ATDC5 cells. These results demonstrated that T-2054 inhibited inflammatory response and promoted extracellular matrix (ECM) synthesis via the NF-κB-signaling pathway (Figure 6a–d).

### 2.7. T-2054 Reduced Cartilage Damage and the Levels of Cytokines in the Serum in DMM-Induced OA Mouse Model

To determine the anti-inflammatory effect of T-2054 on the occurrence and development of OA in vivo, DMM-induced OA mouse models were established. The right knee joints of the mice were cut off and subjected to histopathological examination 10 weeks after destabilization of the medial meniscus (DMM) surgery. The histological analysis was used to examine the activity by H&E and Safranin O fast green staining (SO). The results showed that the infiltration of inflammatory cells was obviously increased, and the cartilage matrix was significantly reduced in the DMM group. However, T-2054 significantly relieved the infiltration of inflammatory cells and degradation of the cartilage matrix compared with the methotrexate (MTX) (Figure 7a–d). At the same time, T-2054 sharply inhibited the release of inflammatory cytokines in the serum. These results demonstrated that T-2054 exerted an anti-inflammatory function in the OA model (Figure 7e–g).

### 2.8. T-2054 Had No Effect on Weight and Major Organs Damage in DMM-Induced OA Mouse Model

Drugs with effective clinical effects and deleterious side effects have been taken seriously in OA treatment [28,29]. Many studies report that organ damage is the major drawback of small molecule drugs. To test the safety of T-2054, the major organs were detected by H&E staining. The data indicated that nearly no damage was found in the lungs, kidney, spleen, liver, and heart, according to the H&E staining slices. These results showed that T-2054 would be safe and have practical value for clinical treatment of OA (Figure 8a,b)

## 3. Discussion

At present, osteoarthritis remains a chronic disease affecting daily activities, particularly among old people, which was mediated by inflammation. Production of inflammatory factors, osteoporosis, trauma, obesity and joint malformation are the main factors in developing OA [30,31]. Acetaminophen and nonsteroidal anti-inflammatory drugs are commonly used clinical drugs to treat OA. However, there have been few effective therapies to withdraw the progress of OA [32]. Recently, a number of anti-inflammatory drugs were applied to the patient. For example, rographolide, licorice, and kaempferol have been reported to possess anti-inflammatory properties [33,34,35,36]. Though these agents could relieve pain and retard the progress of OA, they possess adverse effects, such as gastrointestinal, hepatic, and cardio-renal damage [37]. Hence, anti-inflammatory drugs with effective clinical effects and no deleterious effects have raised interest in the treatment of OA [28,29].

Farnesoid X receptor (FXR) plays an important regulatory role in various inflammatory diseases, such as chronic heart failure, inflammatory bowel disease, and hepatic inflammation [38,39,40]. Obeticholic acid (OCA), a farnesoid X receptor (FXR) agonist, prevented mice from LPS-induced liver injury, ameliorated liver injury induced by carbon tetrachloride via inhibiting inflammatory activity and alleviated the symptoms of acute liver injury [4,17,20].

In the present study, we reported that T-2054, a derivative of OCA, had an anti-osteoarthritis property by screening our small synthetic library of OCA derivatives. Subsequently, we evaluated the effect of T-2054 on the production of proinflammatory cytokines/chemokines (IL-6, IL-8 and TNF-α) and the change of chondrocyte features. IL-6, a proinflammatory cytokine, could lead to the degradation of cartilage and bone; IL-8 is a chemokine recruiting immune cells and leukocyte to pathological tissue; TNF-α, released by activated macrophages, is regarded as a vital proinflammatory cytokine in OA [41,42]. These proinflammatory cytokines further exaggerate chondrocyte apoptosis and decrease the ability to synthesize extracellular matrix (ECM) [10]. Moreover, the production of NO, due to the increased production/activity of inducible nitric oxide synthase (iNOS) and cyclooxygenase-2 (COX-2), results in cartilage tissue damage [43]. In this study, we found that T-2054 obviously reduced the amount of NO in the culture media, the release of proinflammatory cytokines (IL-6, IL-8 and TNF-α), and the expression of iNOS and COX-2 in LPS-induced RAW 264.7 cells. LPS treatment is a common method to simulate an inflammatory reaction in macrophages. In addition, LPS treatment is regarded as an ideal model to test the anti-inflammatory effects of several compounds [44]. Moreover, T-2054 could relieve cartilage degradation and destruction via increasing SOX9 expression and decreasing MMP9 and ADMTS5 expression, which are tightly related to cartilage degradation and destruction in TNF-α-induced ATDC5 cells [14].

In our in vivo studies, T-2054 reduced cytokines’ expression (IL-6, IL-8 and TNF-α) in serum and cartilage degradation and destruction in the OA mice model. The therapeutic effect was far better than that of MTX (2 mg/kg). Moreover, there was no significant difference in the effect on body weight and organ damage after treatment with T-2054, suggesting that T-2054 would be safe and effective for the clinical treatment of OA.

It is reported that transcription factors NF-κB play an important role in the expression of proinflammatory cytokines and cartilage damage, which are the main factors for developing OA [14,45]. In our research, we found the phosphorylation level of P65 was decreased after treated with T-2054 in LPS-activated RAW 264.7 cells and TNF-α-induced ATDC5 cells. These data suggested that T-2054 may act as an inhibitor of NF-κB to suppress inflammatory response and cartilage damage in OA.

## 4. Materials and Methods

The RAW264.7 cell line and ATDC5 cell line were stored in our lab. The cells were cultured in Roswell Park Memorial Institute (RPMI) 1640 medium (RPMI 1640; Gibco Inc., Grand Island, NY, USA) or DMEM/F12 supplemented with 100 U/mL penicillin, 100 μg/mL streptomycin and 10% fetal bovine serum (FBS; Gibco Inc., Grand Island, NY, USA) at 37 °C in 5% CO_2_.

### 4.1. Identification of Nitric Oxide (NO)

To identify the optimal compound, all compounds (0.5 μM) were added into RAW 264.7 cells for 3 h and then incubated with LPS (1 μg/mL) (Sigma-Aldrich; St. Louis, MO, USA) for 24 h. Griess reagent (Sigma-Aldrich; St. Louis, MO, USA) was used to assess the nitrite accumulation in the culture medium at 540 nm with a microplate reader.

### 4.2. Cell Viability Assay

Cell viability was evaluated by Cell-Counting Kit-8 (CCK8) (Sigma-Aldrich; St. Louis, MO, USA)methods. RAW 264.7 cells and ATDC5 cells were seeded into 96-well plates at a density of 8 × 10^3^ cells/well. Different doses of T-2054 (0, 0.1, 1, 10, 100 μM) were added into each well for 24 h. All of the experiments were performed in triplicate.

### 4.3. Real-Time Quantitative PCR

RAW264.7 cells were incubated in LPS (1 μg/mL) for 1 h, and then treated with MTX (Sigma-Aldrich; St. Louis, MO, USA) (0.5 μM) and various doses (0, 0.1 and 0.5 μM) of T-2054 for 6 h. ATDC5 cells were treated with TNF-α (20 ng/mL) or TNF-α and various doses (0, 0.1 and 0.5 μM) of T-2054 for 24 h. The total RNA was collected using Trizol reagent (Invitrogen) followed the manufacturer’s instructions. cDNA was synthesized using the high-capacity cDNA reverse transcription kit (Takara, Dalian, China). For quantitative PCR, all reagents were mixed, including 10 μL of SYBY Green master (Roche Diagnostics GmbH, Mannheim, Germany), 2 μL of primers (10 μM) and 9.5 μL of RNase-free water. Real-time PCR reactions were performed in an Applied Biosystems 7500 real-time PCR system. Reaction cycle conditions were as follows: 95 °C for 10 min of predenaturation conditions, 40 cycles at 95 °C for 15 s, 60 °C for 20 s and 72 °C for 30 s. The *IL-6*, *IL-8*, *TNF-α*, *SOX9*, *MMP9*, *ADAMTS5* and *actin* primers for PCR are shown in Appendix A.

### 4.4. Enzyme-Linked Immunosorbent Assay

RAW 264.7 cells were incubated in LPS (1 μg/mL) for 1 h and then treated with various doses (0, 0.1 and 0.5 μM) of T-2054, and the supernatants were collected after 16 h. The cytokine (IL-6, IL-8 and TNF-α) concentrations in the medium were measured using enzyme-linked immunosorbent assay (ELISA) kits (Pierce Endogen, Rockford, IL, USA) according to the manufacturer’s instructions.

### 4.5. Western Blot

The total lysates were extracted from RAW 264.7 cells and ATDC5 cells using lysis buffer (2% SDS, 10% glycerol, 62.5 mM Tris-HCl buffer, pH 6.8) containing a protease inhibitor cocktail and phosphatase inhibitors (Sigma, Grand Island, NY, USA). Extracts of cells were prepared by using a Bio-Rad protein assay (Bio-Rad, Hercules, CA, USA). Protein samples were subjected to sodium dodecyl sulfate-polyacrylamide gel electrophoresis (10% gel) and transferred onto polyvinylidene difluoride filter (PVDF) membranes (Millipore, Bedford, MA, USA). The membranes were blocked with 5% dried skim milk for 1 h at room temperature and then incubated with primary antibodies against NF-κB (p65), p-p65 and actin overnight at 4 °C. After washing with TBST three times, the membranes were incubated with the secondary antibody at room temperature for 35 min. The band density was measured with a computer-assisted image analysis system (Adobe Systems, San Jose, CA, USA).

### 4.6. DMM-Induced OA Mouse Model and Intraperitoneal Administration of the Test Extract

10-week-old male C57BL/6 mice were established using destabilization of the medial meniscus (DMM) surgery according to a previously described protocol [46]. Animal experiments were approved by the Animal Care and Use Committee of the East China Normal University (protocol m+R 20190301, January 2018). Experimental OA mice were intraperitoneally injected normal control (CON), model (DMM), DMM + PBS, DMM + T-2054 low-dose (1 mg/kg), DMM + T-2054 middle-dose (2 mg/kg), and DMM + T-2054 high-dose (4 mg/kg) or DMM+ methotrexate (MTX) (2 mg/kg) on every other day. After 10 weeks, the mice were sacrificed for histological analysis.

### 4.7. Histological Analysis

The knee joint samples were extracted from mice and fixed in 4% paraformaldehyde followed by decalcification with 10% EDTA solution at 4 °C for two weeks, and then embedding in paraffin blocks. The entire joint was serially selected at a thickness of 5 μm at 40 μm intervals. The deparaffinized sections were stained with Safranin O fast green staining (SO) and hematoxylin and eosin (H&E) staining. The stained sections were photographed digitally using a high-resolution microcomputed tomography specimen scanner.

### 4.8. Detection of Cytokines Levels in Serum

The mice were sacrificed 10 weeks after destabilization of the medial meniscus (DMM) surgery. Blood was obtained from the orbital cavity and then centrifuged at 3000 rpm for 5 min at 4 °C. The serum was separated and used to measure the cytokine (IL-6, IL-8 and TNF-α) concentrations using enzyme-linked immunosorbent assay (ELISA) kits (Pierce Endogen, Rockford, IL, USA) according to the manufacturer’s instructions.

### 4.9. Statistical Analysis

All data were expressed as the mean ± SD. Data were compared using the one-way ANOVA test (SPSS 19.0 software, IBM Corporation, Armonk, NY, USA). The significance of the difference was defined as *p* < 0.05. Each experiment consisted of at least three replicates per condition.

## 5. Conclusions

In conclusion, our study demonstrated that T-2054 significantly suppressed the production of proinflammatory cytokines (IL-6, IL-8 and TNF-α) and the expression of iNOS, COX-2, as well as promoted extracellular matrix (ECM) synthesis via NF-κB pathways. In vitro and in vivo studies confirmed the potential use of T-2054 as an effective therapeutic agent for OA.

## Figures and Tables

**Figure 1 ijms-22-03807-f001:**
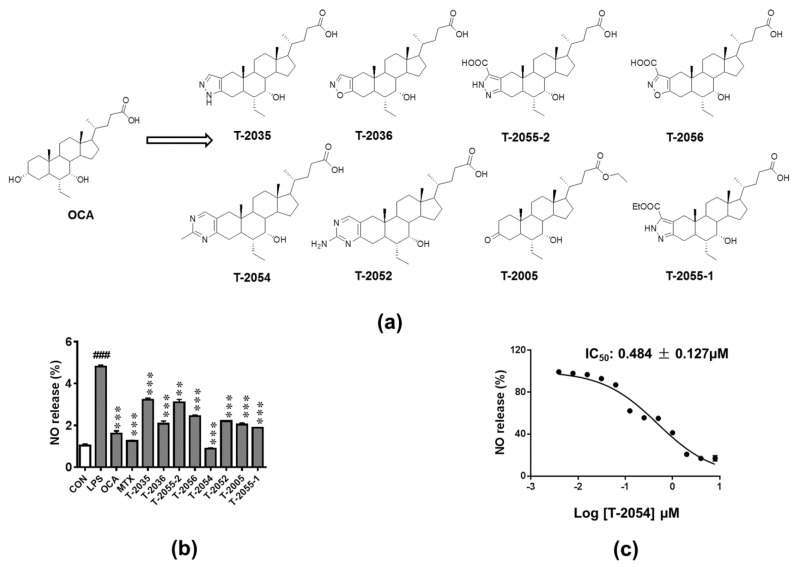
Detect the release of nitric oxide (NO). Structures of obeticholic acid (OCA) and OCA derivatives (**a**). The RAW-264.7 cells were pretreated with compounds and methotrexate (MTX) (0.5 μM) for 3 h before lipopolysaccharides (LPS) stimulation. 24 h later, the amounts of NO in the culture supernatants were measured by Griess agents (**b**). The NO inhibition rates curve of T-2054, with IC_50_ values of 0.484 ± 0.127 μM (**c**). Each value represents the mean ± SD, performed in 3 different experiments. ^##^^#^
*p* < 0.001 vs. normal control (CON); *** *p* < 0.001 vs. LPS group.

**Figure 2 ijms-22-03807-f002:**
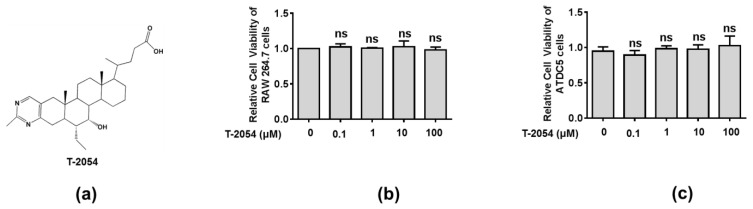
Detect the cell viability of RAW 264.7 cell and ATDC5 cell. Chemical structure of T-2054 (**a**). The RAW 264.7 cell and ATDC5 cell viability were measured by CCK-8 kits after exposure to different concentrations of T-2054 for 24 h (**b**,**c**). Each value represents the mean ± SD, performed in 3 different experiments. ns vs. T-2054 (0 μM).

**Figure 3 ijms-22-03807-f003:**
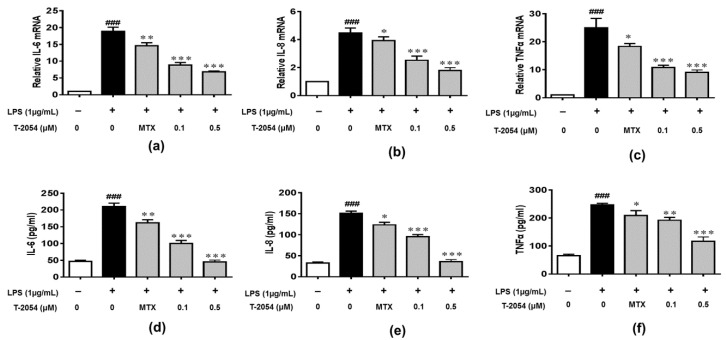
Gene and protein expression of IL-6, IL-8 and TNF-α. The RAW 264.7 cells were treated with LPS (1 μg/mL) 1 h later, T-2054 was added for 6 h. The total RNA was collected, and mRNA levels of *IL-6* (**a**), *IL-8* (**b**) and *TNF-α* (**c**) were analyzed by qPCR. The RAW 264.7 cells were exposed to LPS (1 μg/mL) for 1 h before T-2054 was added. After 16 h, supernatants were collected, and levels of IL-6 (**d**), IL-8 (**e**) and TNF-α (**f**) were analyzed by ELISA. Actin was used as an internal control to normalize the data. Each value represents the mean ± SD, performed in 3 different experiments. ^##^^#^
*p* < 0.001 vs. CON group; * *p* < 0.05; ** *p* < 0.01; *** *p* < 0.001 vs. LPS group.

**Figure 4 ijms-22-03807-f004:**
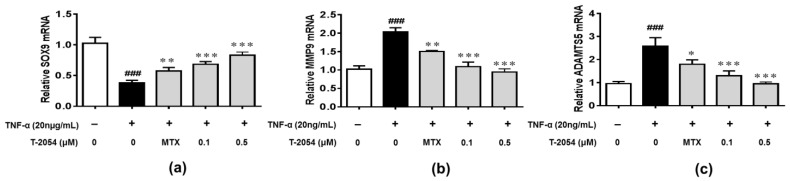
Gene expression of SOX9, MMP9 and ADAMTS5. ATDC5 cells were treated with TNF-α (20 ng/mL) or TNF-α with various doses (0, 0.1 and 0.5 μM) of T-2054 for 24 h. The total RNA was collected and mRNA levels of SOX9 (**a**), MMP9 (**b**) and ADAMTS5 (**c**) were analyzed by qPCR. Actin was used as an internal control to normalize the data. Each value represents the mean ± SD, performed in 3 different experiments. ^##^^#^
*p* < 0.001 vs. CON group; * *p* < 0.05; ** *p* < 0.01; *** *p* < 0.001 vs. TNF-α group.

**Figure 5 ijms-22-03807-f005:**
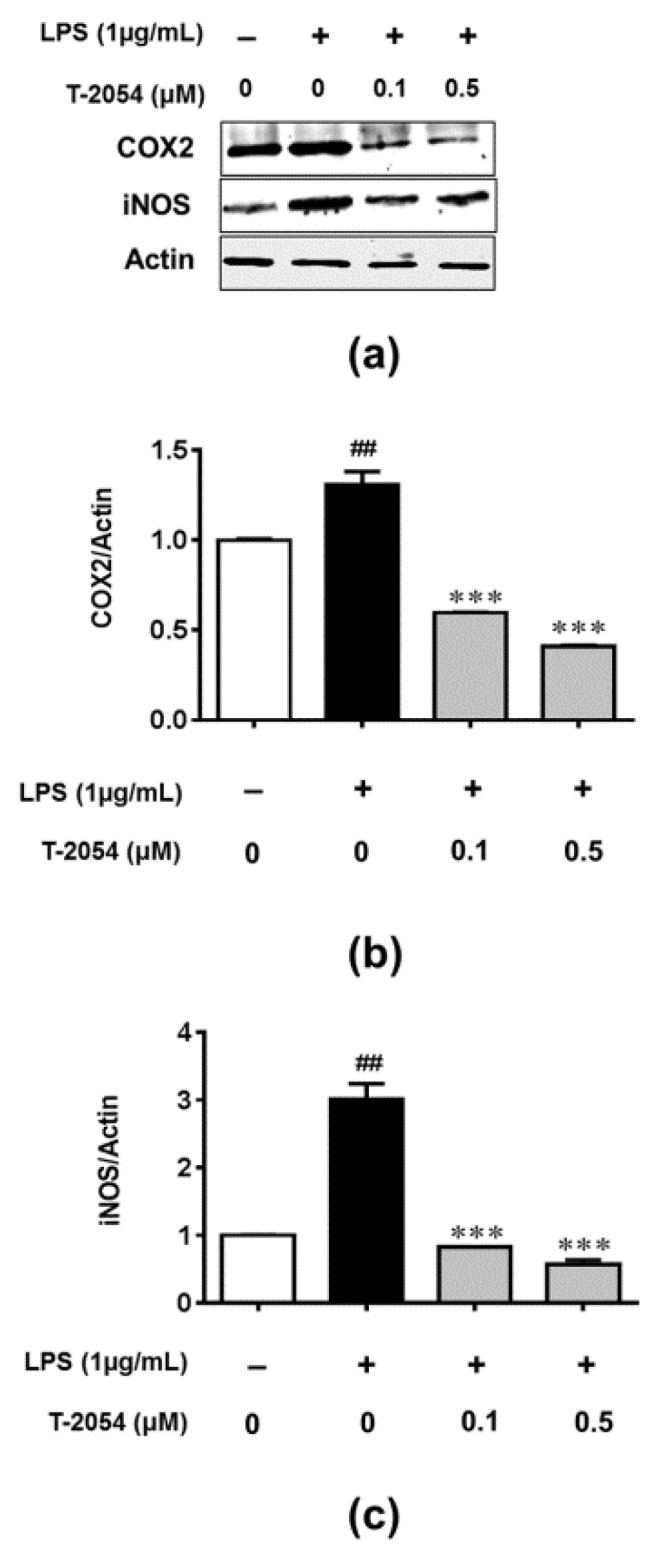
Protein expression of COX-2 and iNOS. The RAW 264.7 cells were treated with LPS (1 μg/mL) 1 h later and treated with or without T-2054 for 3 h, Lysates were prepared, and Western blot was performed using specific COX-2 and iNOS antibodies (**a**). Histogram analysis of the levels of COX-2 (**b**) and iNOS (**c**). Actin was employed as an internal control. Each value represents the mean ± SD, performed in 3 different experiments. ^##^
*p* < 0.01 vs. CON group; *** *p* < 0.001 vs. LPS group.

**Figure 6 ijms-22-03807-f006:**
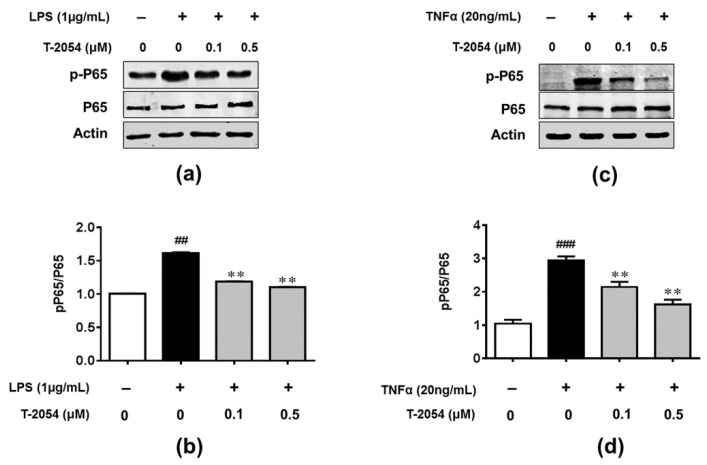
Detected phosphorylation level of NF-κB p65 in RAW 264.7 cells and ATDC5 cells. The RAW 264.7 cells were treated with LPS (1 μg/mL) 1 h later and treated with or without T-2054 for 30 min. Lysates were prepared, and Western blot was performed to detect the expression of p-P65, P65 and actin in LPS-induced RAW 264.7 cells (**a**). Histogram analysis of the levels of p-P65 and P65 (**b**). The ATDC5 cells were treated with TNF-α (20 ng/mL) 1 h later and treated with or without T-2054 for 30 min. Lysates were prepared, and Western blot was performed to detect the expression of p-P65 and P65 in TNF-α-induced ATDC5 cells (**c**). Histogram analysis of the levels of p-P65 and P65 (**d**). Actin was used as an internal control. Each value represents the mean ± SD, performed in 3 different experiments. ^##^
*p* < 0.01, ^##^^#^
*p* < 0.001 vs. CON group; ** *p* < 0.01 vs. Stimulate group.

**Figure 7 ijms-22-03807-f007:**
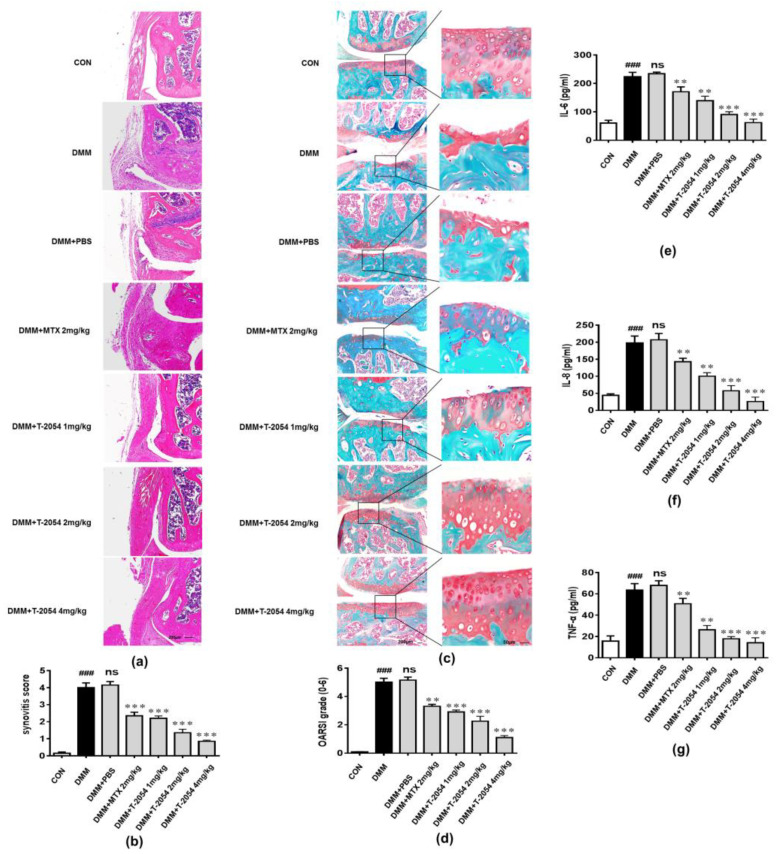
Detected cartilage damage of knee joints. C57BL/6 mice were administered intraperitoneally with PBS, methotrexate (MTX) or T-2054 for 10 weeks after surgery. Representative histopathology of the knee joints from the CON group, destabilization of the medial meniscus (DMM) group, DMM + PBS group, DMM + methotrexate (MTX) (2 mg/kg) group or DMM + T-2054 (1, 2, 4 mg/kg) group were stained with hematoxylin and eosin (H&E) and Safranin O fast green staining (SO) (**a**,**c**). The Osteoarthritis Research Society International (OARSI) scores of each group were presented (**b**). The histopathological scores of each group were assessed to evaluate the infiltration of inflammatory cells and the degree of joint destruction (**b**,**d**). Cytokines production of IL-6 (**e**), IL-8 (**f**) and TNF-α (**g**) was tested in serum by ELISA at the end of the experiment. Each value represents the mean ± SD of 8 mice per group, performed in 2 different experiments. Magnification = 200; bar: 50 μm. ^###^
*p* < 0.001 vs. CON; ns, ** *p* < 0.01 and *** *p* < 0.001 vs. DMM.

**Figure 8 ijms-22-03807-f008:**
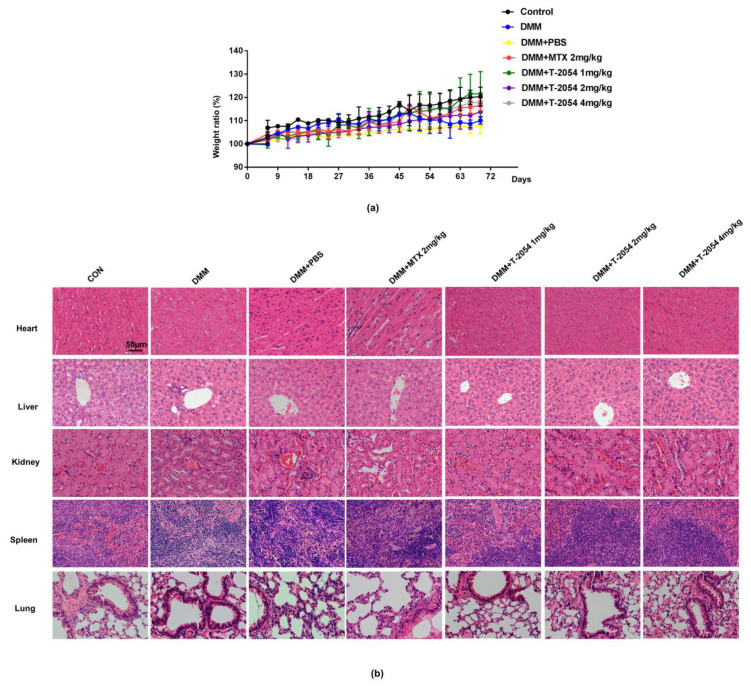
Detect the weight and organ damage in mice. C57BL/6 mice were administered intraperitoneally with PBS, methotrexate (MTX) or T-2054 for 10 weeks after surgery. The representative weight of the mice body was measured (**a**). Representative histopathology of the major organs from the CON group, DMM group, DMM + PBS group, DMM + methotrexate (MTX) (2 mg/kg) group or DMM + T-2054 (1, 2, 4 mg/kg) group were stained with H&E (**b**). Each value represents the mean ± SD of 5 mice per group, performed in 2 different experiments. Magnification = 200; bar: 50 μm.

## Data Availability

The data presented in this study are available on request from the corresponding author.

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
