# Peer review of "Obeticholic Acid Derivative, T-2054 Suppresses Osteoarthritis via Inhibiting NF-κB-Signaling Pathway"

_ijms, 2021, doi:10.3390/ijms22083807_

Round 1

Reviewer 1 Report

The manuscript “Obeticholic acid derivatives, T-2054 suppresses osteoarthritis via inhibiting NF-κB signaling pathway” by Guo et al is a study, in which the effects of OCA and OCA-derivatives are analyzed on inflammatory pathway both in cell culture and in animal model. The topic is very current, but several issues need to be addressed.

Major

The first weakness of this study is the use of Raw 264-7 cell as in vitro model. Those cells are monocytes/macrophages, which have a different embryonic origin compared to chondrocytes. Usually, the Raw- 264-7 cells are used to study the pro-inflammatory pathways, but considering that the cartilage is the main tissue injured by OA, and in cartilage is present only one type of cells, the chondrocytes, it follow that the study of molecules that could be used in OA should be based on the analyses of chondrocytes.

The in vitro studies have to be conducted on chondrocytes or on cells that recapitulate the chondrocyte features.

The second weakness is the use of methotrexate (MTX) as term of comparison for the treatment of OA, but usually, this drug is used to treat the Rheumatoid Arthritis and not the OA. OA is treated with anti-inflammatory drugs or with Disease modifying osteoarthritis drugs (DMOADs), such as glucosamine. So, to compare the effects of T-2054 with MTX is not appropriate, the ability to block the pro-inflammatory cytokine production and the stimulation of extracellular synthesis components have to be compared with a DMOAD.

Minor

The sentence on line 134-135 “after 10-week…..” is probably in the wrong place.

Why the administration of PBS in mice should be weakly detrimental for the mice weight? The graph in fig 7A shows that the yellow line, corresponding to PBD administration, is the lowest one. This point should be discuss.

The primer sequences are shown in Table S1 and not in Table 1.

Some typos throughout the manuscript have to be amended.

Author Response

Point 1: The first weakness of this study is the use of Raw 264-7 cell as in vitro model. Those cells are monocytes/macrophages, which have a different embryonic origin compared to chondrocytes. Usually, the Raw-264-7 cells are used to study the pro-inflammatory pathways, but considering that the cartilage is the main tissue injured by OA, and in cartilage is present only one type of cells, the chondrocytes, it follow that the study of molecules that could be used in OA should be based on the analyses of chondrocytes. The in vitro studies have to be conducted on chondrocytes or on cells that recapitulate the chondrocyte features. 

Response 1: Thanks for this comment. We have detected the effect of T-2054 on ADTC5 chondrocytes, and the results have been added into the manuscript.

Point 2: The second weakness is the use of methotrexate (MTX) as term of comparison for the treatment of OA, but usually, this drug is used to treat the Rheumatoid Arthritis and not the OA. OA is treated with anti-inflammatory drugs or with Disease modifying osteoarthritis drugs (DMOADs), such as glucosamine. So, to compare the effects of T-2054 with MTX is not appropriate, the ability to block the pro-inflammatory cytokine production and the stimulation of extracellular synthesis components have to be compared with a DMOAD.

Response 2: Thanks for this comment. Though methotrexate is usually used to treat the rheumatoid arthritis, it has been utilized in OA patients in recent studies. Therefore, we use methotrexate as term of comparison for the treatment of OA.

Point 3: The sentence on line 134-135 “after 10-week…..” is probably in the wrong place.

Response 3: Thanks for this comment. We have changed the place of “after 10-week…..”.

Point 4: Why the administration of PBS in mice should be weakly detrimental for the mice weight? The graph in fig 7A shows that the yellow line, corresponding to PBD administration, is the lowest one. This point should be discuss.

Response 4: Thanks for this comment. In this study, the mice weight of PBS administration has no significant difference compared with DMM group, which may be due to the effect of surgery on their state.

Point 5: The primer sequences are shown in Table S1 and not in Table 1.

Response 5: Thanks for this comment. We have changed the Table 1 to Table S1.

Point 6: Some typos throughout the manuscript have to be amended.

Response 6: Thanks for this comment. We have amended the typos throughout the manuscript.

Reviewer 2 Report

This manuscript described the Obeticholic acid derivatives T-2054 suppresses osteoarthritis via inhibiting the NF-κB signaling pathway by Dandan Guo et al., which is interesting. However, most mechanism studies of T-2054 was used Raw 264.7 cells. We don't know whether this mechanism is working in chondrocytes. It will be a hamper conclusion. Furthermore, osteoarthritis occurs mainly in anabolic and catabolic imbalance; this manuscript focus on only inflammatory cytokines.

Comments

  1. The authors used macrophage-like RAW 264.7 cells. The authors need to define why the present study used those cell lines, what the correlation is with each other. Strongly suggest using another cell line, such as a chondrocyte.
  2. The authors need to discuss the result of the Nf-Kb pathways analysis in detail.
  3. The authors missed statistical analysis in Figure 1b.
  4. The authors compared None vs. only LPS and LPS vs. LPS+T-2054. Please check all figure legends.
  5. The authors need to correct the "X" axis label of the IC50 value in Figure 1.
  6. DMM surgery described protocol need to add references
  7. In vivo results, authors need to show the "N" number of individuals calculating percentages. It may help to expressed Total Responses.
  8. In figure 6, the figure caption is very confusing. The authors need to revise like below

CON, DMM, DMM+PBS, DMM+MTX 2mg/kg…….

The image's shape is a knee, not an ankle joint.

T-2054 was administered intraperitoneally in figure legends, but T-2054 was administered orally in Materials and methods. Which one is correct?

  1. Please improve the quality of adding the score for synovitis and osteophyte formation in Figure 6.
  2. In Figure 7: the legend expression is wrong.
  3. Please check typos and grammar errors in the paper. I saw some of them.
  4. Please improve the introduction section by adding more recent references.

Author Response

Point 1: The authors used macrophage-like RAW 264.7 cells. The authors need to define why the present study used those cell lines, what the correlation is with each other. Strongly suggest using another cell line, such as a chondrocyte.

Response 1: Thanks for this comment. RAW 264.7 cells are monocytes/macrophages, which are used to study the pro-inflammatory pathways. Besides, we have detected the effect of T-2054 on ADTC5 chondrocytes, and the results have been added into the manuscript.

Point 2: The authors need to discuss the result of the Nf-Kb pathways analysis in detail.

Response 2: Thanks for this comment. We have discussed the result of the Nf-Kb pathways analysis in detail in the section of discussion.

Point 3: The authors missed statistical analysis in Figure 1b.

Response 3: Thanks for this comment. We have added the statistical analysis in Figure 1b.

Point 4: The authors compared None vs. only LPS and LPS vs. LPS+T-2054. Please check all figure legends.

Response 4: Thanks for this comment. We have checked all figure legends.

Point 5: The authors need to correct the "X" axis label of the IC50 value in Figure 1.

Response 5: Thanks for this comment. We have corrected the "X" axis label of the IC50 value in Figure 1.

Point 6: DMM surgery described protocol need to add references.

Response 6: Thanks for this comment. We have added the reference into the section of references.

Point 7: In vivo results, authors need to show the "N" number of individuals calculating percentages. It may help to expressed Total Responses

Response 7: Thanks for this comment. We have showed the number in vivo results.

Point 8: In figure 6, the figure caption is very confusing. The authors need to revise like below

CON, DMM, DMM+PBS, DMM+MTX 2mg/kg…….

The image's shape is a knee, not an ankle joint.

T-2054 was administered intraperitoneally in figure legends, but T-2054 was administered orally in Materials and methods. Which one is correct?

Response 8: Thanks for this comment. We have revised the caption in figure 6, and changed ankle joint to knee joint. "T-2054 was administered intraperitoneally" is correct and we have amended in the manuscript.

Point 9: Please improve the quality of adding the score for synovitis and osteophyte formation in Figure 6.

Response 9: Thanks for this comment. We have increased the synovitis score in Figure 6.

Point 10: In Figure 7: the legend expression is wrong.

Response 10: Thanks for this comment. We have corrected the legend expression in Figure 7.

Point 11: Please check typos and grammar errors in the paper. I saw some of them.

Response 11: Thanks for this comment. We have amended the typos throughout the manuscript.

Point 12: Please improve the introduction section by adding more recent references.

Response 12: Thanks for this comment. We have improved the introduction section.

Reviewer 3 Report

The authors investigated the effect of a compound, T-2054, which belongs to the obeticholic acid family, on a macrophage cell line stimulated with LPS as a model of OA.  In addition, a mouse OA (DMM) model is performed and the effect on cartilage analysed. To corelate with the macrophage in vitro model the synovial membrane should also be adressed using a synovitis score. Neither the abstract nor introduction or discussion sections provide the information that a macrophage cell line was used.  What is the rationale to use exactly this cell model in combination with LPS here and not another? In regard to OA this is a very seldom used model which needs justification and discussion. The last paragraph of the introduction might be suitable to explain this. NFkappaB is involved in many signaling pathways important for healthy cell performance and hence, there is a risk of side effects.The histopathology is limited to inflammatory cells I guess that there might be some degenerative changes which have not been checked such as a hyperemia in the kidneys treated with treated with the extract, the loss of structure of the glomerula, however, the low magnification and limite quality of images does not allow a deeper insight.

In regard to the mouse model the headline (4.6.) states „oral administration of the test extract“ but in the paragraph I can find: „mice were intraperitoneally injection“ what is wrong here? Only mice are mentioned in the abstract but not the kind of model…At all, there lack crucial information in the method section.

The abstract is insufficiently structured and some information as mentioned above is absent.

Please mention the LPS stimulation, explain all abbreviations, add concentrations and time points of investigation,

„most cause“ write „most common cause“

line 24: „degeneration of cartilage matrix“: what was exactly observed? „effectively relieved“ write „relieve“

some abbreviations should be introduced/be explained (NO, IL)

„relieved the release of…“ mention the cell type used to show this

Introduction

Line 34: „progressive of“ the „of“ seems to be surplus

Line 58: „binds affinity“ add „with high“

Line 65, 71 and other sites: the font size changes

The discussion is short and remains superficial.

Title: „derivatives“ please write singular since the study is restricted to one of them

Legend of figure 1: explain the abbreviations used, do it generally in every legend! mention the statistical tests used in each legend and the number of independent experiments performed.

„CTL“ might mean control? In the other figures it is abreviated as CON. I would prefer as more commonly used to write „co“ consistently throughout the manuscript.

2.2

Line 85: „optical concentration“ does it mean „optimal“?

Figure 3: the legend needs a heading. „Gene and protein expression of …..“ The first sentence here already describes the experimental procedure.

Line 103: „actin“ which type?

Line 112-113: „activated“…“activated“ within one sentence… improve style

Figure 6: „MTX“ explain abbreviation

Line 149: „serum by ELISA“ at which time point?

Line 153: „and deleterious side effects have raised interest…“ sounds confusing

Line 158: „for clinical“ what? please complete it.

Figure 7: label the axes oft he diagram (days, weight?)

Histology: show a higher magnification

„the infiltration of inflammatory cells“ how about degenerative changes? I am worried by the histopathology of the hearts treated with 2054. The quality of the images is not good but it seems to differ substantially from the control and also PBS being possibly degenerated.

Line 172: „and deleterious side effects“ does it mean „no deleterious…“?

Line 180: write „are mainly caused“

Lines 191-193: sentence is doubled

Figure 8: the contrast is weak and hence the labeling difficult to read

Author Response

Point 1: In regard to the mouse model the headline (4.6.) states „oral administration of the test extract“ but in the paragraph I can find: „mice were intraperitoneally injection“ what is wrong here? Only mice are mentioned in the abstract but not the kind of model…At all, there lack crucial information in the method section

Response 1: Thanks for this comment. In this study, we use intraperitoneal administration to treat mice, and we have corrected in the manuscript. We have added crucial information into the method section.

Point 2: The abstract is insufficiently structured and some information as mentioned above is absent.

Response 2: Thanks for this comment. We have added crucial information into abstract.

Point 3: Please mention the LPS stimulation, explain all abbreviations, add concentrations and time points of investigation,

Response 3: Thanks for this comment. We have added explain of all abbreviations, concentrations and time points into the manuscript.

Point 4: „most cause“ write „most common cause“

Response 4: Thanks for this comment. We have changed "most cause" into "most common cause".

Point 5: line 24: „degeneration of cartilage matrix“: what was exactly observed? „effectively relieved“ write „relieve“

Response 5: Thanks for this comment. We have changed the "effectively relieved" into relieve.

Point 6: some abbreviations should be introduced/be explained (NO, IL)

Response 6: Thanks for this comment. We have added the abbreviations into the section of abbreviations.

Point 7: „relieved the release of…“ mention the cell type used to show this

Response 7: Thanks for this comment. We have showed this.

Point 8: Line 34: „progressive of“ the „of“ seems to be surplus

Response 8: Thanks for this comment. We have deleted the "of".

Point 9: Line 58: „binds affinity“ add „with high“

Response 9: Thanks for this comment. We have added with high.

Point 10: Line 65, 71 and other sites: the font size changes

Response 10: Thanks for this comment. We have corrected the font size.

Point 11: The discussion is short and remains superficial.

Response 11: Thanks for this comment. We have discussed our results in more depth in the section of discussion.

Point 12: Title: „derivatives“ please write singular since the study is restricted to one of them.

Response 12: Thanks for this comment. We have changed "derivatives" to   "derivative".

Point 13: Legend of figure 1: explain the abbreviations used, do it generally in every legend! mention the statistical tests used in each legend and the number of independent experiments performed.

Response 13: Thanks for this comment. We have explained the abbreviations, and added the statistical tests and the number of independent experiments in every legend.

Point 14: „CTL“ might mean control? In the other figures it is abreviated as CON. I would prefer as more commonly used to write „co“ consistently throughout the manuscript.

Response 14: Thanks for this comment. We have changed the "CTL" to "CON" in the manuscript.

Point 15: Line 85: „optical concentration“ does it mean „optimal“?.

Response 15: Thanks for this comment. It means "optimal" and we have corrected it.

Point 16: Figure 3: the legend needs a heading. „Gene and protein expression of …..“ The first sentence here already describes the experimental procedure.

Response 16: Thanks for this comment. We have added a heading into the legend.

Point 17: Line 103: „actin“ which type?

Response 17: Thanks for this comment. We have changed the "actin" to "Actin" in the manuscript.

Point 18: Line 112-113: „activated“…“activated“ within one sentence… improve style

Response 18: Thanks for this comment. We have improved style.

Point 19: Figure 6: „MTX“ explain abbreviation

Response 19: Thanks for this comment. We have explained the abbreviation of "MTX" in figure 6.

Point 20: Line 149: „serum by ELISA“ at which time point?

Response 20: Thanks for this comment. The serum was collected 10 weeks after destabilization of the medial meniscus (DMM) surgery.

Point 21: Line 153: „and deleterious side effects have raised interest…“ sounds confusing

Response 21: Thanks for this comment. We have checked it.

Point 22: Line 158: „for clinical“ what? please complete it.

Response 22: Thanks for this comment. We have completed it.

Point 23: Figure 7: label the axes of the diagram (days, weight?)

Response 23: Thanks for this comment. We have added the diagram into figure 7.

Point 24: Histology: show a higher magnification

Response 24: Thanks for this comment. We have shown a higher magnification of histology.

Point 25: „the infiltration of inflammatory cells“ how about degenerative changes? I am worried by the histopathology of the hearts treated with 2054. The quality of the images is not good but it seems to differ substantially from the control and also PBS being possibly degenerated.

Response 25: Thanks for this comment. We have shown a higher magnification of histology.

Point 26: Line 172: „and deleterious side effects“ does it mean „no deleterious…“?

Response 26: Thanks for this comment. It means "no deleterious", and we have changed it.

Point 27: Line 180: write „are mainly caused“

Response 27: Thanks for this comment. We have changed it.

Point 28: Lines 191-193: sentence is doubled

Response 28: Thanks for this comment. We have deleted it.

Point 29: Figure 8: the contrast is weak and hence the labeling difficult to read

Response 29: Thanks for this comment. We have changed it.

Round 2

Reviewer 1 Report

The authors responded to some issues raised during the first revision, however some changes are still need.

The authors must add a reference regarding the use of methotrexate for treatment of OA.

A revision of the English language throughout the text is still required, mainly in the new added parts (yellow sentences).

In Fig 3 the mRNA expression and the protein production of the different cytokines should be aligned, IL-6 mRNA with IL-6 proteins and Il-8 mRNA with IL-8 proteins.

At lane 243, why the authors say “obviously” in the following sentence “…..that T-2054 obviously reduced the amount of NO …”?

Author Response

Response to Reviewer 1 Comments

Point 1: The authors must add a reference regarding the use of methotrexate for treatment of OA. 

Response 1: Thanks for this comment. We have added a reference regarding the use of methotrexate for treatment of OA.

Point 2: A revision of the English language throughout the text is still required, mainly in the new added parts (yellow sentences).

Response 2: Thanks for this comment. We have revised the English language throughout the text.

Point 3: In Fig 3 the mRNA expression and the protein production of the different cytokines should be aligned, IL-6 mRNA with IL-6 proteins and Il-8 mRNA with IL-8 proteins.

Response 3: Thanks for this comment. We have aligned the mRNA expression and the protein production of the different cytokines.

Point 4: At lane 243, why the authors say “obviously” in the following sentence “…..that T-2054 obviously reduced the amount of NO …”?

Response 4: Thanks for this comment. We use“obviously” because the amount of NO in the culture media, the release of pro-inflammatory cytokines (IL-6, IL-8 and TNF-α), and the expression of iNOS and COX-2 were significantly decreased.

Reviewer 2 Report

This manuscript's quality much improved. The authors addressed all my concerns. 

Author Response

Thanks for your comments that has greatly improved my article and made it more perfect.

Reviewer 3 Report

There are still many flaws/mistakes in the novel text. Several sentences make no sense and should be rewritten.

examples are given below.

I think "ATDC5 cells" (chondrogenic cells of mouse teratucarcinoma) is the correct name
"expression protein and mRNA levels": please write mRNA and protein expression levels. I
line 25 "chondrocyte vitality" write only vitality since chondrocytes follows later in the same sentence
line 31: the coma after "as well as" might be surplus
line 52 include..."that" seems to be surplus
Line 53: the degradation... This sentence makes no sense, please rewrite
line 57: "activation of cytokines and NO" how are these cytokines activated? They are only released. I think it makes no sense, please rewrite this sentence according to the cited references!
line 70: "binds affinity with high" means "binds with high affinity"
line 97 and line 123 and 174 and line 197 and 212: "experimental" means "experiments" or "experimental settings"
line 160: "of" seems to be surplus, line 164: "the" can be omitted before "similar"
line 174: insert a blank before "Actin"
line 187: use the same font size for Methotrexate
line 194: please write correctly OARSI score
line 201: instead of "find" better to write "reporting"
line 208: write "organ damage"
line 217: the coma is possibly surplus in the novel text
line 219: "withdraw" correct font size/type?
line 239: write "exaggerate", next line "in the" should be omitted
line 241-242: iNOS etc please check the content of the sentence/citation - makes no sense
line 254: "safety" should be "safe"
line 259: "in our research..." sentence makes no sense - rewrite
line 286: insert blank,metotrexate was already abbreviated before

Author Response

Response to Reviewer 3 Comments

Point 1: I think "ATDC5 cells" (chondrogenic cells of mouse teratucarcinoma) is the correct name.

Response 1: Thanks for this comment. We have corrected the cell name.

Point 2: "expression protein and mRNA levels": please write mRNA and protein expression levels.

Response 2: Thanks for this comment. We have changed "expression protein and mRNA levels" into "mRNA and protein expression levels".

Point 3: line 25 "chondrocyte vitality" write only vitality since chondrocytes follows later in the same sentence

Response 3: Thanks for this comment. We have changed "chondrocyte vitality" into" vitality".

Point 4: line 31: the coma after "as well as" might be surplus

Response 4: Thanks for this comment. We have deleted it.

Point 5: line 52 include..."that" seems to be surplus

Response 5: Thanks for this comment. We have deleted it.

Point 6: Line 53: the degradation... This sentence makes no sense, please rewrite

Response 6: Thanks for this comment. We have rewritten the sentence.

Point 7: line 57: "activation of cytokines and NO" how are these cytokines activated? They are only released. I think it makes no sense, please rewrite this sentence according to the cited references!

Response 7: Thanks for this comment. We have rewritten the sentence.

Point 8: line 70: "binds affinity with high" means "binds with high affinity"

Response 8: Thanks for this comment. We have changed it.

Point 9: line 97 and line 123 and 174 and line 197 and 212: "experimental" means "experiments" or "experimental settings"

Response 9: Thanks for this comment. We have changed "experimental" into "experiments".

Point 10: line 160: "of" seems to be surplus, line 164: "the" can be omitted before "similar"

Response 10: Thanks for this comment. We have deleted it.

Point 11: line 174: insert a blank before "Actin".

Response 11: Thanks for this comment. We have inserted a blank before "Actin".

Point 12: line 187: use the same font size for Methotrexate.

Response 12: Thanks for this comment. We have changed it.

Point 13: line 194: please write correctly OARSI score.

Response 13: Thanks for this comment. We have corrected it.

Point 14: line 201: instead of "find" better to write "reporting".

Response 14: Thanks for this comment. We have changed "find "into" reporting".

Point 15: line 208: write "organ damage"

Response 15: Thanks for this comment. We have changed "major organs damage "into" organ damage ".

Point 16: line 217: the coma is possibly surplus in the novel text

Response 16: Thanks for this comment. We have deleted it.

Point 17: line 219: "withdraw" correct font size/type?

Response 17: Thanks for this comment. We have changed it.

Point 18: line 239: write "exaggerate", next line "in the" should be omitted

Response 18: Thanks for this comment. We have changed it.

Point 19: line 241-242: iNOS etc please check the content of the sentence/citation - makes no sense

Response 19: Thanks for this comment. We have checked it.

Point 20: line 254: "safety" should be "safe"

Response 20: Thanks for this comment. We have changed "safety" into "safe".

Point 21: line 259: "in our research..." sentence makes no sense - rewrite

Response 21: Thanks for this comment. We have rewritten it.

Point 22: line 286: insert blank, metotrexate was already abbreviated before

Response 22: Thanks for this comment. We have inserted blank.
